# Reversing the charge transfer between platinum and sulfur-doped carbon support for electrocatalytic hydrogen evolution

Qiang-Qiang Yan [1,4], Dao-Xiong Wu [1,4], Sheng-Qi Chu[2], Zhi-Qin Chen[1], Yue Lin [1], Ming-Xi Chen [1], Jing Zhang[2], Xiao-Jun Wu [1,3]* & Hai-Wei Liang [1]*

Metal–support interaction is of great significance for catalysis as it can induce charge transfer between metal and support, tame electronic structure of supported metals, impact adsorption energy of reaction intermediates, and eventually change the catalytic performance. Here, we report the metal size-dependent charge transfer reversal, that is, electrons transfer from platinum single atoms to sulfur-doped carbons and the carbon supports conversely donate electrons to Pt when their size is expanded to ~1.5 nm cluster. The electron-enriched Pt nanoclusters are far more active than electron-deficient Pt single atoms for catalyzing hydrogen evolution reaction, exhibiting only 11 mV overpotential at 10 mA cm$^{-2}$ and a high mass activity of 26.1 A mg$^{-1}$ at 20 mV, which is 38 times greater than that of commercial Pt/C. Our work manifests that the manipulation of metal size-dependent charge transfer between metal and support opens new avenues for developing high-active catalysts.

[1] Hefei National Laboratory for Physical Sciences at the Microscale, School of Chemistry and Materials Sciences, University of Science and Technology of China, 230026 Hefei, China. [2] Beijing Synchrotron Radiation Facility, Institute of High Energy Physics, Chinese Academy of Sciences, 100049 Beijing, China. [3] Synergetic Innovation of Quantum Information & Quantum Technology, CAS Key Laboratory of Materials for Energy Conversion, and CAS Center for Excellence in Nanoscience, University of Science and Technology of China, 230026 Hefei, Anhui, China. [4] These authors contributed equally: Qiang-Qiang Yan, Dao-Xiong Wu. *email: xjwu@ustc.edu.cn; hwliang@ustc.edu.cn

Supported nanoparticle catalysts play a crucial role in modern chemistry industry. It has been demonstrated in literature that the support, more than a carrier to finely disperse and stabilize metal, would interplay with supported metal and influence catalytic reactivity and selectivity[1]. Classic strong metal–support interaction (SMSI) was proposed and termed first by Tauster et al. in 1978, involving the migration of oxide supports and encapsulation of metal particles during high-temperature reductive treatment, which effectively suppress CO adsorption[1]. After decades of research, SMSI has been intensively investigated and covered many kinds of interaction mechanisms including oxide encapsulation[2,3], charge transfer[4–7], synergistic reaction in catalytic process[8–10] as well as anti-sintering of nanoparticles[11].

Charge transfer, as a typical SMSI that occurs at the interface between metal nanoparticle and support, can introduce electronic modification on metal particles and directly influence their adsorption behavior during catalytic process, which will eventually change their catalytic reactivity and selectivity[4,5,12–15]. The importance of charge transfer and electronic perturbations on reactivity was manifested recently by Rodriguez and coworkers, in which Pt nanoparticles contacting with ceria support showed enhanced ability to dissociate O–H bonds in water and finally achieved the best catalytic activity at the greatest electronic perturbation[4]. Such an electronic perturbation, arising from charge transfer between metal and support, was termed as electronic metal–support interaction (EMSI) by Campbell[5]. Many works have demonstrated that the charge transfer was relevant to the properties of supports, such as the reducibility[6,7,16]. Meanwhile, it was also reported that the preparation method[17] could influence the d-band centers of supported metals. Although these studies concerning the effects of supports and treatment methods on charge transfer boost our understanding of EMSI, much less attentions have been focused on charge transfer tuned by metal itself, especially the particle size[7,14,18,19].

As the diminishment of fossil fuel and deterioration of environment, hydrogen has become the focus of research and aroused interests of scientists around the world for its cleaning and reproduction[20]. Hydrogen is generally produced by water electrolysis, photocatalytic water splitting, and steam reforming of methane[21–23]. Among these technologies, water electrolysis transforms electrical energy to chemical energy directly with the merits of high efficiency, green and pure production, emerging a nice prospect for commercialization. Hydrogen evolution reaction (HER) is a vital reaction in water electrolysis and thus, looking for active, stable, and low cost catalyst to drive the HER and cut the cost for commercialization is significantly important and highly desired. Platinum-group metal (PGM) are the best catalysts for HER, however, the high price and the low abundance restrict their wide usage. The grand challenge is therefore to develop low cost PGM-free electrocatalysts[24,25], or to greatly decrease the Pt loading by enhancing the specific activity of Pt or improving the utilization efficiency of Pt atoms[26–29].

Herein, we report the metal size-dependent charge transfer based on the mesoporous sulfur-doped carbon (S–C) supported Pt single atom and nanocluster catalysts (Fig. 1). By spectroscopic characterizations and density functional theory (DFT) calculations, we find that the S–C support captures electrons from Pt single atoms via the strong chemical Pt–S interaction, while the electron transfer is converse from S–C to Pt when the Pt size is increased to nanocluster. The electron-enriched Pt nanocluster catalysts show much higher catalytic activity for HER than electron-deficient Pt single atom catalysts as well as commercial Pt/C catalyst.

## Results

### Synthesis and structural characterizations.
The mesoporous S–C supports with high sulfur content of >12 wt% and high specific surface area of >1000 $m^2\,g^{-1}$ were prepared by carbonization of molecular precursors with silica nanoparticles as templates (Supplementary Figs. 1 and 2), according to our previously reported works[24,30]. X-ray photoelectron spectroscopy (XPS) analyses revealed that the doped sulfur in S–C existed primarily in the form of C–S–C, together with a small amount of S–$O_x$ species (Supplementary Fig. 3). With S–C as supports, we prepared Pt single atom (PtSA/S–C) and Pt nanocluster (PtNC/S–C) catalysts with 5 wt% Pt loading via wet-impregnation of chloroplatinic acid on S–C followed by $H_2$-reduction at 300 and 700 °C, respectively (Fig. 1). XPS and temperature-programmed reduction measurements confirmed the complete removal of chlorine and the reduction of $H_2PtCl_6$ before 200 °C in $H_2$ atmosphere (Supplementary Figs. 4 and 5).

Aberration-corrected high-angle annular dark field-scanning transmission electron microscopy (HAADF-STEM) was first performed to reveal the atomic-resolution structure of the Pt/S–C catalysts. For the PtSA/S–C catalyst prepared at 300 °C, we observed numerous atomically dispersed Pt species as well as a few loose ensembles of Pt atoms on the S–C support (Fig. 2a). When the $H_2$-reduction temperature was increased to 700 °C, nearly all Pt species exited as Pt nanoclusters with an average size of 1.56 nm, which were homogeneously dispersed on the S–C support for the PtNC/S–C catalyst (Fig. 2b and Supplementary Fig. 6). Further HAADF-STEM observations showed that the Pt nanoclusters in PtNC/S–C were crystalline and mainly enclosed by (111) and (200) crystal planes of fcc Pt (Fig. 2c). X-ray diffraction (XRD) analyses of the catalysts confirmed the absence of crystalline Pt in PtSA/S–C and the emerging of fcc Pt phase in PtNC/S–C (Fig. 2d). Energy-dispersive X-ray spectrometry (EDS) elemental mapping revealed that the S-moieties in S–C were thermally stable and still distributed homogeneously over the carbon support after loading of Pt at 700 °C (Supplementary Fig. 7).

### Spectroscopic characterizations.
We then measured X-ray absorption fine structure (XAFS) and XPS to characterize the coordination environments and the electronic structures of Pt species in the catalysts. The normalized X-ray absorption near edge structure (XANES) present in Supplementary Fig. 8 revealed that the white line intensities of both Pt/S–C catalysts were located between Pt foil and $PtO_2$, indicating the partially oxidized Pt species in these catalysts. The k space of $k^2$-weighted Pt $L_3$-edge in the Supplementary Fig. 9 shows only minor noise, suggesting the high data quality for all samples. The $k^2$-weighted Fourier transform of extended X-ray adsorption fine structure (EXAFS) of Pt $L_3$-edge of the Pt/S–C catalysts, as well as reference samples were shown in Fig. 3a. Clearly, PtSA/S–C exhibited only a peak near 1.8 Å without any Pt–Pt contribution between 2 and 3 Å, confirming the atomic dispersion of Pt atoms over the S–C support. We therefore presumed that the few ensembles of Pt atoms observed by HAADF-STEM in PtSA/S–C come from loosely packed Pt atoms or the overlapping of Pt atoms in 3-D porous structures of S–C[31,32]. The peak of PtSA/S–C at 1.8 Å in EXAFS could be ascribed to the Pt–S coordination by comparing with the EXAFS of $PtS_2$ and $PtO_2$. In contrast, PtNC/S–C showed an additional peak at 2.6 Å, close to the Pt–Pt contribution of Pt foil, suggesting the coexistence of Pt–Pt bonds along with the Pt–S bonds in the PtNC/S–C catalyst.

To visually explore the coordination conditions of Pt, wavelet transform (WT) of the $k^2$-weighted EXAFS spectra was obtained in Fig. 3b, which can directly reflect the structure information in the resolution of R space and k space. The WT intensity maximum of PtSA/S–C and $PtS_2$ both occurred near R space of 1.7–1.8 Å and k space of 6 Å$^{-1}$, confirming the similar

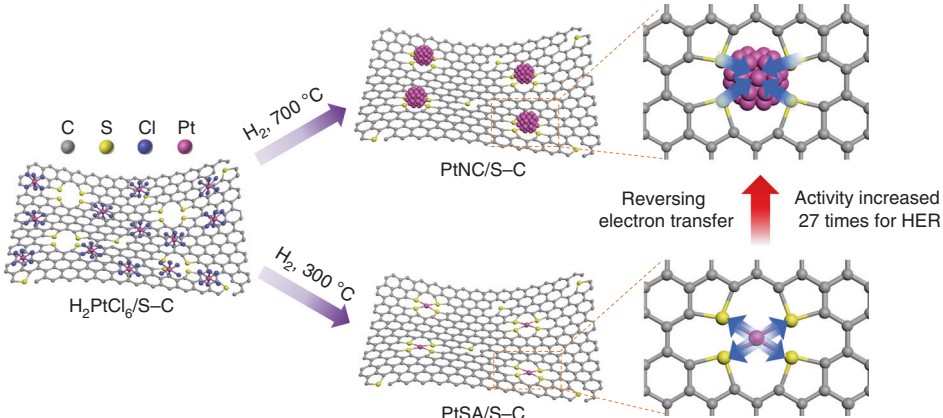

**Fig. 1** Schematic model of the catalyst preparation process and metal size-dependent charge transfer. PtSA/S–C and PtNC/S-C were prepared by wet-impregnation of chloroplatinic acid on the S–C supports followed by $H_2$-reduction at 300 and 700 °C, respectively. When Pt size increases from single-atom to nanocluster, the electron transfer direction is reversed, resulting in greatly enhanced activity for catalyzing HER

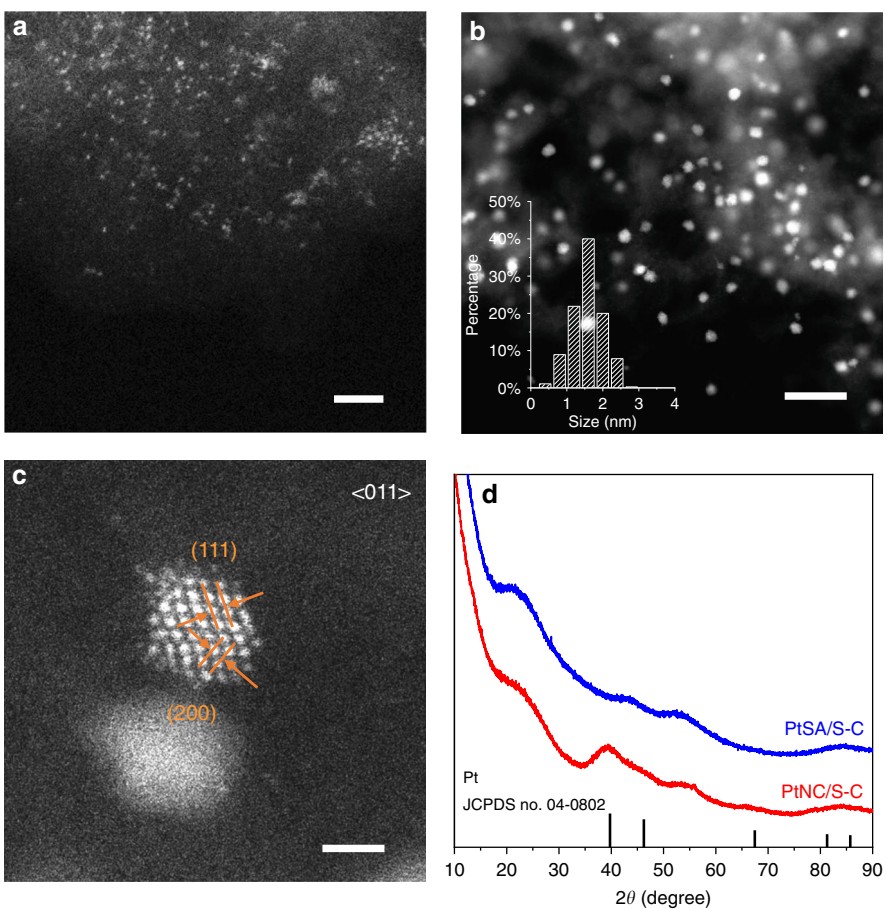

**Fig. 2** Structural characterization of PtSA/S-C and PtNC/S-C. **a** HAADF-STEM image of PtSA/S-C. Scale bar, 2 nm. **b** HAADF-STEM image of PtNC/S-C. Scale bar, 10 nm. **c** High-resolution HAADF-STEM image of PtNC/S-C. Scale bar, 1 nm. **d** XRD patterns of PtSA/S-C and PtNC/S-C. The standard peaks of Pt are shown in black lines

coordination structure of Pt–S bonds in the first coordination shell. As for PtNC/S-C, besides the Pt–S bond, a new WT intensity maximum near 2.8 Å and 9 Å$^{-1}$ appeared, which was associated with Pt–Pt bonding. Further quantitative analysis of the coordination structure was conducted by the fitting of $R$ space of Pt/S-C (Supplementary Fig. 10 and Supplementary Table 1). The coordination number of Pt–S bonding of PtSA/S-C in the first coordination shell was estimated to be 3.2, close to the coordination number of 3.5 in PtS$_2$, implying the similar coordination conditions of Pt–S$_4$ in the first coordination shell[33]. Due to the aggregation of Pt atoms in PtNC/S-C, the average coordination number of Pt–S bonding decreased while that of Pt–Pt bonding increased.

XPS was then investigated to directly probe the surface charge state of Pt (Fig. 3c). Clearly, PtSA/S-C showed a high binding energy of about 72.5 eV in Pt $4f$ spectra, rather close to the

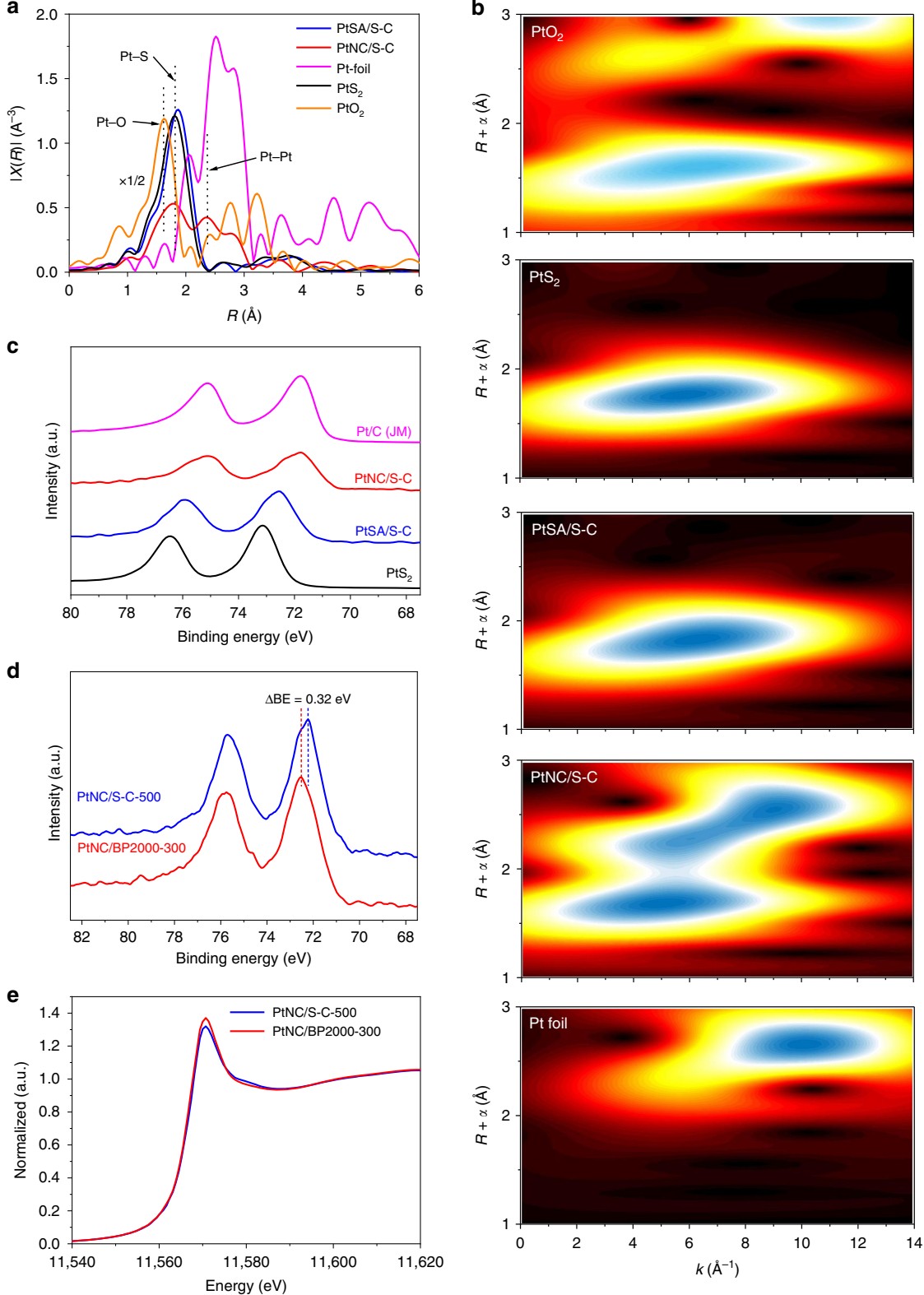

**Fig. 3** Spectroscopic characterizations of the Pt/S–C catalysts. **a** Fourier transform of $k^2$-weighted Pt L$_3$-edge of PtSA/S–C, PtNC/S–C, PtO$_2$, PtS$_2$, and Pt foil. **b** Wavelet transform for the $k^2$-weighted EXAFS spectra of PtSA/S–C, PtNC/S–C, PtO$_2$, PtS$_2$, and Pt foil. **c** High-resolution XPS spectra (Pt 4$f$) of PtSA/S–C, PtNC/S–C, and PtS$_2$. **d** High-resolution XPS spectra (Pt 4$f$) of PtNC/BP2000-300 and PtNC/S–C-500. **e** Normalized XANES spectra at the Pt L$_3$-edge of PtNC/BP2000-300 and PtNC/S–C-500

binding energy of $PtS_2$ (73.1 eV), revealing the same nature of high oxidation state. Such high oxidation state of Pt in PtSA/S–C suggested the electron transfer from Pt atoms to S–C via the interfacial Pt–S bonds as revealed by above XAFS analyses. Compared to PtSA/S–C, the PtNC/S–C catalyst exhibited a negatively shifted binding energy of 71.8 eV. It is rather interesting that the binding energy of PtNC/S–C is similar to that of Pt/C, although the particle size of latter is larger, which may suggest that electron transfer reversely from the S–C support to Pt clusters. To rule out the possible influence of particle size-induced change of the binding energy shift[34], we prepared the Pt nanoclusters supported on high-surface-area carbon black BP2000 support (PtNC/BP2000-300) with the similar Pt size as the ones supported on S–C (PtNC/S–C-500) and then compared their Pt 4f binding energy (Supplementary Figs. 11 and 12, Supplementary Table 2). We found that the PtNC/S–C-500 catalyst displayed a lower Pt 4f binding energy than PtNC/BP2000-300 (Fig. 3d). Moreover, the white line intensity of PtNC/S–C-500 in XANES was also lower than that of PtNC/BP2000-300 (Fig. 3e). These results definitely manifest the reversed electron transfer from S–C to Pt nanoclusters.

**DFT calculations of metal size-dependent charge transfer**. To further understand the electron transfer between Pt and S–C support, DFT calculations were performed to analyze the difference charge density. As discussed above, due to the similar first coordination shell between PtSA/S–C and $PtS_2$, the motif of Pt-$S_4$ was imbedded in the lattice of graphene. Considered the computational consumption and experimental size of Pt nanocluster, here, $Pt_{38}$ cluster with truncated octahedron shape is chosen as model (Fig. 4a and Supplementary Fig. 13), which expose both (111) and (200) surfaces and is the same to PtNC/S–C as revealed by the high-resolution HAADF-STEM image (Fig. 2c). The binding energy calculations confirmed the stable configuration of $Pt_{38}$, which is also consistent with the previously reported calculation results[35]. We investigated the charge transfer between S-Graphene and a series of $Pt_n$ cluster with atom number ranging from 1 to 44 by the difference charge analysis (Supplementary Fig. 14). The structures of $Pt_1$/S-Graphene and $Pt_{38}$/S-Graphene are fully optimized, as shown in Supplementary Fig. 15. Notably, the charge transfer direction is distinct between Pt single atom and nanocluster ($n > 1$). In particular, about 0.069 electrons are transferred from Pt atom to S-Graphene in $Pt_1$/S-Graphene, and conversely, 0.657 electrons are transferred to $Pt_{38}$ from S-Graphene in $Pt_{38}$/S-Graphene (Fig. 4). The electron affinity analyses and d-band center results further suggested the charge transfer reversal between Pt and S-Graphene when the Pt atom

number increased from 1 to 44, which is due to the increase of electron affinity and the downshift of d-band center that can enhance the electron capture ability of Pt clusters (Supplementary Figs. 16 and 17). These computational analyses are well consistent with experiment results, indicating the metal-size-dependent charge transfer reversal between Pt and S–C. Note that Tessonnier et al. studied the charge distribution of different Pd particles ranging from 38 atoms to 239 atoms on carbon support by DFT calculations[19]. In that work, the direction of charge transfer between Pd and carbon supports was found to be independent of particle size.

**Electrocatalysis performance**. The HER activities of Pt/S–C catalysts were then evaluated using a three-electrode setup with graphite rod as the counter electrode in $N_2$-saturated 0.5 M $H_2SO_4$ at room temperature. Since the binding energy of reaction intermediate H* during HER in acidic media dominates the reactivity, it is rather meaningful to explore the electronic structure–activity relationship[36]. The PtSA/S–C catalyst with atomically dispersed Pt atoms showed a relatively poor HER activity with an overpotential of 53 mV at 10 mA $cm^{-2}$. In sharp contrast, the PtNC/S–C catalyst exhibited a much higher activity with an overpotential of only 11 mV at 10 mA $cm^{-2}$, which was also significantly superior to the commercial Pt/C catalyst (Fig. 5a, Supplementary Figs. 18 and 19). To further quantitatively compare the HER activity, the mass activity of the catalysts at the overpotential of 20 mV was calculated after being normalized to the Pt loading and the results are shown in Fig. 5b. The mass activity of PtNC/S–C was high up to 26.1 A $mg^{-1}$, which was 27 times and 38 times greater than that of PtSA/S–C (0.964 A $mg^{-1}$) and commercial Pt/C (0.684 A $mg^{-1}$), respectively. Also, the activity comparison of PtNC/S–C with recently reported noble metal electrocatalysts in terms of the overpotential at 10 mA $cm^{-2}$ and loading amount clearly indicated that PtNC/S–C exhibited almost the smallest loading and the best HER activity simultaneously (Fig. 5c), making it being one of the best HER catalysts to our knowledge (Supplementary Table 3). To investigate intrinsic reaction kinetics of Pt/S–C for HER, Tafel slope was plotted and fitted (Fig. 5d). PtNC/S–C displayed a Tafel slope of 23.51 mV $dec^{-1}$, similar to the commercial Pt/C catalyst (29.01 mV $dec^{-1}$), suggesting that both of them share the same rate-determining step, where two hydrogen intermediates desorb and form $H_2$ molecular (Tafel step)[28]. Differently, PtSA/S–C showed a high Tafel slope of 46.92 mV $dec^{-1}$, indicating the slow reaction kinetics of the Pt–S single sites. Moreover, PtNC/S–C exhibited remarkable exchange current density as high as 1.732 mA $cm^{-2}$ (Fig. 5e), which is nearly two times larger than PtSA/S–C (0.921 mA $cm^{-2}$). The Nyquist plots show a much smaller charge transfer resistance of PtNC/S–C than that of PtSA/S–C, demonstrating the rapid HER kinetics (Supplementary Fig. 20). Our results are in strong contrast with the previous reports that single Pt atom catalyst could catalyze HER efficiently with low overpotential and Tafel slope[26,28,29,37–41], implying that the chemical environments and electronic structures of Pt single atom may play a determinant role in the catalysis[39,42]. We found that the PtNC/S–C-500 showed a higher HER activity than BP2000 supported platinum catalyst (PtNC/BP2000-300) with similar particle size but a lower electron density (Supplementary Fig. 21), suggesting that the enhanced HER activity could be associated to the enriched electron state of Pt cluster on the S–C supports.

The accelerated degradation test (ADT) was performed to explore the stability of the catalysts in $N_2$-saturated 0.5 M $H_2SO_4$. After 10,000 cyclic voltammetry cycles in the range of −0.15 to +0.15 V (vs. RHE), PtNC/S–C showed a negligible loss of activity

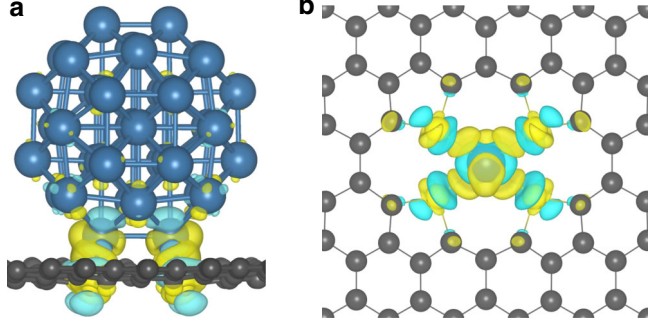

**Fig. 4** Electronic structure analysis of Pt/S-Graphene. **a**, **b** Difference charge density analysis of $Pt_{38}$/S-Graphene **a** and $Pt_1$/S-Graphene system **b**. Differential charge density with yellow and cyan colors represent positive and negative electron density isosurfaces, respectively. The value of isosurface is 0.003e/bohr$^3$

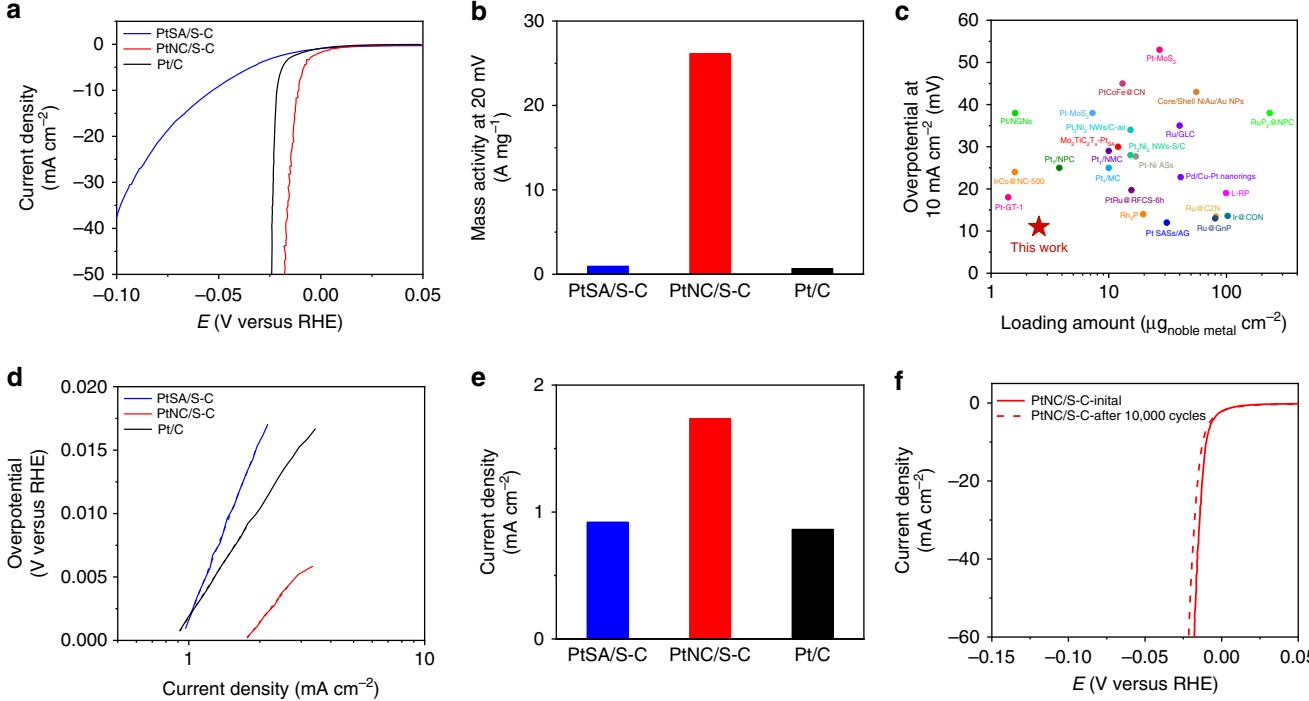

**Fig. 5** Electrocatalytic performance of PtSA/S-C and PtNC/S-C. **a** HER polarization curves of PtSA/S-C, PtNC/S-C, and commercial Pt/C in 0.5 M H$_2$SO$_4$ at room temperature. **b** Mass activity of PtSA/S-C, PtNC/S-C, and commercial Pt/C at an overpotential of 20 mV (vs. RHE). **c** Activity comparison of PtNC/S-C with other recently reported noble metal electrocatalysts in terms of loading amount and the overpotential at 10 mA cm$^{-2}$ (seen Supplementary Table 2 for more details for the comparison). **d** Tafel slope of PtSA/S-C, PtNC/S-C, and commercial Pt/C acquired from **a**. **e** Exchange current density $j_0$ of PtSA/S-C, PtNC/S-C, and commercial Pt/C. **f** Stability test of PtNC/S-C. The polarization curves were recorded initially and after 10,000 potential cycles between −0.15 and +0.15 V (vs. RHE) at 100 mV s$^{-1}$ in 0.5 M H$_2$SO$_4$. All the results of our catalysts are IR corrected

with a negative shift only about 4 mV at 60 mA cm$^{-2}$, indicating the outstanding stability of the PtNC/S–C catalyst (Fig. 5f). HAADF STEM images of the PtNC/S–C catalyst after ADT proved that the average size of Pt nanocluster was 1.6 nm, approaching that of pristine PtNC/S–C (Supplementary Fig. 22). EDS elemental mapping images also suggested that S are still distributed homogeneously over the carbon after ADT (Supplementary Fig. 23).

## Discussion

To explore the origin of HER activity and elucidate its correlation with the electronic interaction between Pt and S–C, DFT calculations were performed. First, the possible adsorption sites (S1–S6) of hydrogen atom on surface of Pt slab and cluster were considered, as illustrated in Fig. 6a, b. The calculated absorption energies of hydrogen atom on Pt(111) surface were −0.56, −0.52, −0.53, and −0.57 eV for S1–S4 sites, respectively, agreeing well with reported results[43]. The corresponding hydrogen absorption free energy ΔG was −0.32, −0.28, −0.29, and −0.33 eV, respectively. The comparison of free energies of hydrogen atom on Pt slab, Pt$_{38}$, and Pt$_{38}$/S-Graphene are illustrated in Fig. 6d. On the Pt$_1$/S-Graphene, the calculated free energy of hydrogen was 1.08 eV, implying that the single Pt atom loaded on S-Graphene exhibits low catalytic activity toward HER. Differently, the calculated free energy of hydrogen on Pt$_{38}$/S-Graphene was as low as −0.07 eV on S5 site of Pt$_{38}$ cluster, implying the superior catalytic activity of Pt$_{38}$/S-Graphene over the Pt(111) surface and Pt$_1$/S-Graphene. All the hydrogen absorption configuration of Pt$_{38}$, Pt$_1$/S-Graphene, and Pt$_{38}$/S-Graphene were summarized in Supplementary Figs. 24 and 25. Note that the best Gibbs free energy was −0.27 eV on S6 site in free-standing Pt$_{38}$ and was closed to that of Pt(111). However, the free energy dramatically reduced to

−0.07 eV if Pt$_{38}$ was supported on S-doped graphene, suggesting the pivotal role of the S-Graphene substrate in regulating the electronic structures of Pt and the HER performance (Fig. 6d).

To uncover the effect of electronic structure on the HER activity of Pt$_1$/S-Graphene and Pt$_{38}$/S-Graphene, the charge analysis results (Δe) were correlated to the change of calculated absorption energies (δG) of hydrogen atom (Fig. 6c). Twelve hydrogen adsorption sites were tested, denoted as site 1 to site 12, respectively. We found that the δG of 12 hydrogen adsorption sites with different charge were different. We speculate that the δG is affected by the charge of Pt atoms near the corresponding adsorption site. To verify this issue, electrons and holes were introduced into Pt(111) slab, respectively. The HER catalytic activity of all four absorption sites (S1–S4) were enhanced when Pt(111) were negatively charged and the activity change trend was converse when Pt(111) were positively charged (Fig. 6d). Moreover, the detailed pCOHP analyses show that the charge transfer results in an increase in Pt–H bonding interaction at some adsorption sites (δG < 0) and a decrease in bonding interaction at some other adsorption sites (δG > 0) (Supplementary Fig. 26). It is worth noting that besides the electronic structure, the geometrical effect induced by the SMSI would also affect the catalytic activity (Supplementary Table 4). Although it is challenging to completely rule out the influence of geometric effect on the catalytic activity, it is safe to conclude that the outstanding HER performance of Pt-NC/S-C at some hydrogen absorption sites arise from the electron-enriched state of Pt, as a result of the size-dependent charge transfer between Pt and the S–C support.

In summary, we demonstrate that electronic SMSI between Pt and the S–C supports could be manipulated by changing the Pt morphologies from single atoms to nanoclusters. In contrast to the electron transfer from single Pt atoms to the S–C support, the charge transfer direction was converse from S–C to Pt when the

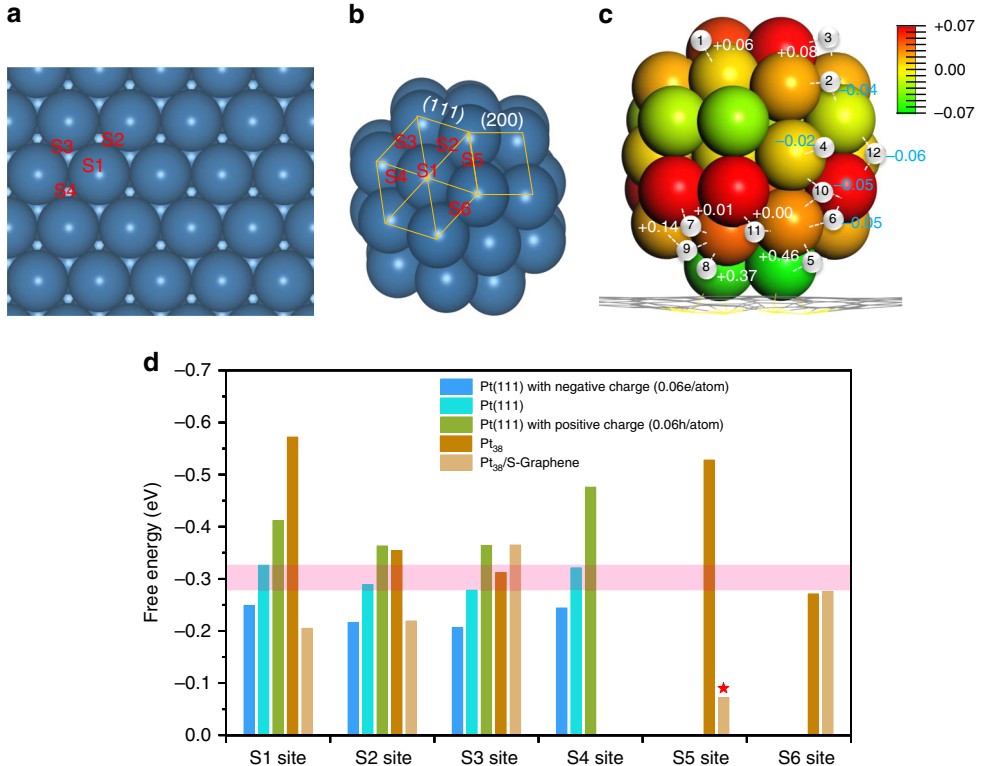

**Fig. 6** Hydrogen atoms absorption sites and catalytic free energies. **a, b** Hydrogen atom absorption sites of Pt(111) **a** and Pt$_{38}$ cluster **b**. S1–S4 denote the top site, bridge site, hcp site, and fcc site on crystal face (111), respectively. S5 denotes the bridge site at the edge of (111) and (200), and S6 is the bridge site at the edge of adjacent (111) face. **c** Difference bader charge analysis and difference $\Delta G$ energy analysis of Pt$_{38}$/S-Graphene. The value of $\Delta e$ in Pt atom can be read from the color bar. The small white balls with number mark the absorption site with the corresponding $\delta G$. **d** Calculated free energy diagram of Pt(111), Pt(111) with negative charge (0.06e/atom), Pt(111) with positive charge (0.06hole/atom), Pt$_{38}$ and Pt$_{38}$/S-Graphene. The translucent pink belt denotes the cover range of $\Delta G$ of the Pt(111)

particle size was increased to nanocluster. We found that the electron-enriched Pt nanoclusters show much higher catalytic activity for HER than the Pt single atoms and commercial Pt/C catalyst. Such metal size-dependent charge transfer phenomenon extends the understanding of SMSI and provides a new strategy to modify the electronic structure of metal nanocatalysts for various catalysis processes.

## Methods

**Synthesis of the S–C supports.** Mesoporous S–C supports were fabricated by the cobalt-assisted pyrolysis of molecular precursors with silica nanoparticles as templates[24,30]. First, 2.0 g 2,2′-bithiophene (J&K chemicals) and 1.0 g Co (NO$_3$)$_2$·6H$_2$O (Sinopharm Chemical Reagent Co. Ltd.) were dissolved in 120 mL tetrahydrofuran (Sinopharm Chemical Reagent Co. Ltd.) before 2.0 g SiO$_2$ nanoparticles (7 nm, Sigma-Aldrich S5130) were added under vigorous stirring to form a homogeneous solution. Then the solution was dried by a rotary evaporator and grinded to yield a mixed powder, which was subjected to pyrolysis at 800 °C (5 °C min$^{-1}$) under N$_2$ atmosphere for 2 h. Finally, the pyrolyzed product was etched successively in 2.0 M NaOH solution for 3 days at room temperature and in 0.5 M H$_2$SO$_4$ for 8 h at 90 °C to remove SiO$_2$ templates and cobalt, respectively. Inductively coupled plasma-atomic emission spectrometer (ICP-AES) measurement indicated that the left cobalt content in S–C was negligible (<0.1 wt%).

**Synthesis of PtSA/S–C and PtNC/S–C.** The PtSA/S–C and PtNC/S–C catalysts were synthesized by conventional wet-impregnation method. Briefly, 50 mg S–C was well dispersed in 30 mL deionized water containing 7.0 mg H$_2$PtCl$_6$ by ultrasonic treatment. Then the H$_2$PtCl$_6$/S–C solution was dried by using a rotary evaporator. Afterwards, the dried powder was subjected to tube furnace and reduced at 300 and 700 °C (5 °C min$^{-1}$), respectively, under flowing Ar/H$_2$ for 2 h to produce the PtSA/S–C and PtNC/S–C catalysts.

**Characterization.** X-ray powder diffraction was performed with Japan Rigaku DMax-γA rotation anode x-ray diffractometer using Cu K-α radiation. The operation voltage and current were 40 kV and 150 mA. The scan speed was set at 1.2° min$^{-1}$ with 2θ range of 10–90°. TEM analyses was conducted by a JEOL-2010F transmission electron microscope with an acceleration voltage of 200 kV. The aberration-corrected HAADF-STEM images were obtained using JEM ARM200F TEM operated at 200 kV. EDS mapping images were carried out on FEI Talos F200X, equipped with Super X-EDS system (four systematically arranged windowless silicon drift detectors) at 200 kV. ICP-AES measurement was carried out using Optima 7300 DV. XPS measurements were carried out on a VG ESCALAB MK II X-ray photoelectron spectrometer with an exciting source of Mg K$_\alpha$ = 1253.6 eV.

**XAFS experiments and data analyses.** XAFS data of Pt L$_3$-edge were acquired at the 1W1B beamline of the Beijing Synchrotron Radiation Facility operated at 2.5 GeV and 200 mA and at the BL14W1 beam line of Shanghai Synchrotron Radiation Facility (SSRF) operated at 3.5 GeV and 220 mA. The raw data analyses were conducted by using Athena program in IFEFFIT software package. The energy was first calibrated, then the pre-edge background of spectrum was subtracted and post-edge was normalized. $k^2$-weighted EXAFS oscillations ranging from 2.5 to 12.2 Å were Fourier transformed to obtain a radial distribution function. The data fitting was carried out using Artemis program in IFEFFIT[44]. The passive electron factors, $S_0^2$, were first determined by fixing Pt–Pt coordination number (CN) of experimental Pt foil data to be 12 and then were fixed to analyze other samples.

**Electrochemical measurement.** Electrochemical measurement was conducted by using Pine Instruments in a three-electrode electrochemical cell. Ag/AgCl was used as the reference electrode and graphite rod instead of Pt wire was chosen as the counter electrode to avoid the dissolution and re-deposition of Pt on the working electrode. A rotating disk glassy-carbon with diameter of 5 mm (Pine Instruments) was used as working electrode. The Ag/AgCl reference electrode was calibrated in H$_2$-saturated 0.5 M H$_2$SO$_4$ and potentials in this work are with respect to RHE.

2.0 mg catalysts was dispersed in 2.0 mL isopropanol containing 40 μL 5 wt% Nafion solution (Sigma-Aldrich) and then stirred for 1 day and sonicated for 2 h to get a homogeneous ink. Afterwards, a certain amount of ink was dropped on the working electrode (2.55 μg$_{Pt}$ cm$^{-2}$) and dried under ambient temperature. For

comparison, the ink of commercial 20 wt% Pt/C (Johnson Matthey) was also prepared and deposited on the working electrode (10.2 µg$_{Pt}$ cm$^{-2}$). Before measurement, the catalysts were cycled, about 50 cycles between −0.25 and +1.0 V (vs. Ag/AgCl) to maximize the activity. Linear sweep voltammetry was taken from −0.15 to +0.05 V (vs. RHE) in N$_2$-saturated 0.5 M H$_2$SO$_4$ at the scan rate of 2 mV s$^{-1}$. The rotating speed of working electrode was set at 1600 r.p.m. to remove H$_2$ gas bubble during the test. For the ADT, cyclic voltammetry was cycled between −0.15 and +0.15 V (vs. RHE) with the scan rate of 100 mV s$^{-1}$ for 10,000 cycles. After that, the catalysts were cycled again between −0.25 and +1.0 V (vs. Ag/AgCl) for about 100 cycles before LSV measurement. The Nyquist plot of Pt/S–C and commercial Pt/C were obtained using Zahner work station at a potential of −0.24 V (vs. Ag/AgCl) over a frequency range of 0.1 Hz–100,000 Hz.

Electrochemical impedance spectroscopy (EIS) measurement was performed by utilizing CHI760 electrochemical workstation (Chenhua) to obtain solution resistance. The frequency range is between 0.01 and 100,000 Hz and initial voltage and amplitude voltage are set at 0 and 0.005 V (vs. Ag/AgCl), respectively.

Tafel slopes were obtained by linear fitting the plot derived from logarithm of current density vs. overpotential. The Tafel slopes were determined from Tafel equation:

$$\eta = b \log j + c \qquad (1)$$

where $b$ is the Tafel slope, $\eta$ is the overpotential, $j$ is the current density, $c$ is the intercept.

The exchange current density was obtained by formula[27]:

$$j_0 = e^{(-2.303c/b)} \qquad (2)$$

**Computational details**. Spin-polarized DFT simulations were performed with the Vienna ab initio simulation package (VASP)[45,46]. The generalized gradient approximation (GGA) of the Perdew–Burke–Ernzerhof (PBE) functional and the projector augmented-wave (PAW) potential were employed[46–48]. An energy cutoff of 500 eV was used for the plane-wave expansion of the electronic wave function. All structures were fully relaxed until the maximum force on atom was <0.05 eV Å$^{-1}$ and the convergence criterion of self-consistent calculation was 10$^{-5}$ eV. Van der Waals correction was included by using the DFT-D3 method[49]. The thickness of vacuum layer was >12 Å.

To investigate the HER activity, we built models of Pt(111) slab and S-doped graphene supported Pt cluster (Pt$_n$/S-Graphene, where $n$ is the number of Pt atoms). A seven layers $2 \times 2 \times 1$ supercells of Pt(111) slab with the atoms in the bottom three layers fixed at their bulk positions was used. A set of $7 \times 7 \times 1$ k-points was sampled by using Monkhorst–Pack scheme[50]. Supercells of $6 \times 6$ and $8 \times 8$ graphene unitcells were used to model the Pt$_1$/S-Graphene and Pt$_{38}$/S-Graphene, respectively. A set of $2 \times 2 \times 1$ k-points were sampled by using gamma-centered Monkhorst–Pack scheme to describe the Brillouin zone.

The Gibbs free energy ($\Delta G$) of the adsorption of hydrogen atom was calculated according to Eq. (1):

$$\Delta G = \Delta E_H + \Delta E_{ZPE} - T\Delta S_H \qquad (3)$$

where $\Delta E_H$, $\Delta E_{ZPE}$, and $\Delta S_H$ represent the hydrogen absorption energy, the correction of zero-point energy and the entropy difference between the absorbed hydrogen atom (H$^*$) and free H$_2$ molecule, respectively. $T$ is the temperature, which was chosen as 298.15 K. $\Delta E_H$ was calculated according to Eq. (2):

$$\Delta E_H = E_{total} - E_{primitive} - E_{H_2}/2 \qquad (4)$$

where $E_{total}$ is the total energy of the system absorbed with hydrogen atom, $E_{primitive}$ is the total energy of the system without absorbing hydrogen atom, and $E_{H_2}$ is the energy of H$_2$ molecule. $\Delta E_{ZPE} - T\Delta S_H$ is about 0.24 eV[51–54]. Moreover, the difference in free energy of hydrogen absorption site is defined as $\delta G = \Delta G(Pt_{38}/S$-Graphene) – $\Delta G(Pt_{38})$, where $\Delta G(Pt_{38}/S$-Graphene) and $\Delta G(Pt_{38})$ denote the free energy of the absorption of sites Pt$_{38}$/S-Graphene and Pt$_{38}$, respectively.

To uncover the effect of S–C on the HER activity of Pt$_{38}$/S-Graphene, bader charge analysis and difference charge density analysis were performed. The difference in charge density is defined as $d\rho = \rho(system) - \rho(substrate) - \rho(Pt_n)$, where $\rho(system)$ denotes the charge density of the whole system, $\rho(substrate)$ denotes the charge density of S-doped carbon substrate removing Pt$_n$ without relaxation, and $\rho(Pt_n)$ denotes the charge density of Pt$_n$ removing substrate and keeping the Pt atoms frozen. The difference in bader charge of each Pt atom is calculated according to $\Delta e = e(Pt_{38}/S$-Graphene) $- e(Pt_{38})$, where $e(Pt_{38}/S$-Graphene) and $e(Pt_{38})$ are the charge of each Pt atom in Pt$_{38}$/S-Graphene and fully relaxed Pt$_{38}$, respectively.

## Data availability

All data presented in this study and the codes for DFT calculations are available from the corresponding authors (H.-W.L. and X.-J.W.) upon request.

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

## Acknowledgements

H.-W.L. acknowledges the funding supports from the NSFC (Grant 21671184), the Fundamental Research Funds for the Central Universities (Grant WK2340000076), and the Joint Funds from Hefei National Synchrotron Radiation Laboratory (Grant KY2060000107), USTC start-up funding, and the Recruitment Program of Thousand Youth Talents. X.-J.W. acknowledges the funding supports from the MOST (2016YFA0200602, 2018YFA0208603), NSFC (51521001, 21890751), Anhui Initiative in Quantum Information Technology, CAS Interdisciplinary Innovation Team, and the Fundamental Research Funds for the Central Universities. We acknowledges Beijing Synchrotron Radiation Facility (1W1B beam line) and Shanghai Synchrotron Radiation Facility (BL14W1 beam line) for the synchrotron beam time.

## Author contributions

H.-W.L., X.-J.W., Q.-Q.Y. and D.-X.W. conceived and designed the project. Q.-Q.Y. and Z.-Q.C. synthesized and characterized the catalysts. X.-J.W. and D.-X.W. conducted the DFT calculations. Y.L. performed the HAADF-STEM characterization. J.Z., S.-Q.C. and M.-X.C. performed the X-ray absorption spectra measurements and analyzed the data. Q.-Q.Y., D.-X.W., H.-W.L. and X.-J.W. co-wrote the manuscript. All authors discussed the results and commented on the manuscript.

## Competing interests

The authors declare no competing interests.
