## [Peer Review File · Nature Communications]

Reviewer #1 (Remarks to the Author):

In the manuscript, the authors show that the charge transfer between Pt nanoclusters and mesoporous sulfur-doped graphene (S-C) can be reversed through the size of the Pt nanocluster. The authors showed experimental results based on XPS and theoretical results based on DFT to confirm this. It has also been shown that the catalytic activity for hydrogen evolution reaction varies with the size of these nanoclusters. However, there are some issues that need clarification.

(1) The shape chosen for the Pt₃₈ nanocluster model needs to be clarified. In the paper [Barnard et al., ACS Nano 2009, 3, 6, 1431], it has been shown that the shape of gold nanoparticles depends on temperature and size. For the similar temperature and size of the manuscript, the most stable format would be icosahedral. Although this result is for gold nanoparticles, it is expected that platinum nanoparticles will be similar.

Response: Many thanks for the reviewer's comment on this issue. Barnard et al. depicted the phase diagram of Au clusters in <T, D> (T denotes the temperature, D denotes the critical diameter) space based on the theory and experiment analyses (ACS nano, 2009, 3, 1431-1436.). The phase diagram predicted the structure order of the icosahedral < decahedral < fcc structures. Clusters of many elements favor the icosahedral growth either beginning from a small size range or the icosahedral structures become favorable with an increase in size such as for Au. However, the growth behavior commonly found in many elements is unlikely to happen in Pt element. For example, Kumar et al. showed that, for the Pt element, the bulk structure is favored ranging from a relatively small size range of about 40 atoms (also even for some smaller clusters) to quite large clusters, which is different from Au (Physical Review B, 2008, 77, 205418.). Moreover, the 3x3x4 cuboid is the most stable configuration for Pt₃₆ cluster; and the most stable configuration of Pt_n cluster gradually transform from cuboid into regular octahedron from n=36 to n=44.

The binding energy of a series of Pt clusters are showed in Fig. R1a. Our results

indicate that the most stable configuration of Pt₃₈ cluster is Pt₃₆ cuboid capped with two Pt atoms on the 3x4 plane (Fig. R1b). The binding energy of Pt₃₈ with the shape of truncated octahedron (Fig. R1c) is very close to that of cuboid-shape Pt₃₈, which means the former is also energy favored. These results are consistent with the Kumar's analyses (Physical Review B, 2008, 77, 205418). Additionally, the HAADF-STEM image results show that the exposed crystal faces of PtNC/S-C prepared experimentally are (111) and (200) (Fig. R1d), which is in good agreement with the atomic array and exposed crystal plane of the proposed Pt₃₈ structure. Overall, considering the computational and experimental results, we chose the Pt₃₈ clusters with truncated octahedron shape in the current simulation work.

Parts of above theory results and discussion have been added in the revised manuscript (page 7) and Supplementary Information (Fig. S13).

Figure R1. (a) Binding energy of a series of Pt_n cluster. The binding energy of cuboid-shape Pt₃₈ and truncated octahedron Pt₃₈ are marked with blue and red colors, respectively. The binding energy of Pt_n cluster is defined as $E_b = E(\text{system})/n$, where n is the number of Pt atom. (b) Cuboid-shape Pt₃₈ cluster. (c) Truncated octahedron Pt₃₈ cluster with the zone axis [011] perpendicular to the paper. The red and yellow lines mark the crystal face (111) and (200) of Pt₃₈ cluster, respectively. (d) HAADF-STEM image of PtNC/S-C.

(2) On the charge transfer, is there any change in the work function with the size change of Pt nanocluster? If there is no significant change, why does single atom platinum transfer electrons to the S-C? In this case, maybe the crystal field splitting of the platinum d-orbitals can shift in a manner that induces such charge transfer. I would recommend a density of state analysis for this question.

Response: Thanks for the reviewer's valuable comments here. As far as we know, the work function is usually applied for crystal materials and is related to the corresponding exposed crystal face. The Pt cluster used in this work is rather small ($<10 \text{ \AA}$). We therefore believe that the use of electron affinity (denoted as EA, $EA = E(\text{Pt}_n) - E(\text{Pt}_n^-)$) is more suitable than work function to evaluate the ability of Pt clusters to capture charges. We calculated the electron affinity of a series of Pt clusters from Pt_1 to Pt_{44} (Fig. R2a). We found that the EA of Pt cluster increased slightly with the cluster size, which means that the ability of the Pt cluster system to capture electrons is enhanced.

As suggested by the reviewer, we further study the charge transfer by the density of state analyses for the Pt_1 and Pt_{38} (Fig. R3). The d-orbital of Pt_1 is distributed between -1 and 0.5 eV, while the d-orbital of Pt_{38} clusters is clearly split with a wide distribution from -6 to +0.5 eV. The d-band center is used here to evaluate the degree of displacement of the d-orbit as the cluster size increases (Fig. R2b). The d-band center of the Pt single atom is shallow, and its spin-down d-orbital in PDOS (Fig. R3a) is even located at the Fermi level. These d-orbital electrons should be easily to interact with the substrates. From Pt_1 to Pt_6 , the d-band center decreases rapidly. The d-band center of large Pt_n cluster ($n>6$) is much more negative than that of Pt_1 , indicating that the d-orbital of Pt_n cluster moves to a deeper level relative to Pt_1 . The d-band center shift may be associated to the charge transfer from substrate to Pt cluster. The binding energy of a series of Pt clusters are showed in Fig. R2c. As the size increases, the Pt cluster with deeper d-band center becomes more stable with more stable electronic structure, and may further leading to the larger EA. Therefore, it can be concluded that the charge transfer between Pt and S-C is highly relevant to the platinum particle size, which is

induced by the change of electron affinity as well as the shift of d-band center.

The above theory results and discussion have been added in the revised manuscript (page 7) and Supplementary Information (Figs S16 and S17).

Figure R2. EA (a), d-band center (b) and binding energy (c) of a series of Pt_n cluster. The bind energy of cuboid-shape Pt₃₈ and truncated octahedron Pt₃₈ are marked with blue and red color, respectively. Bind energy of Pt_n cluster is defined as $E_b = E(\text{system})/n$, where n is the number of Pt atom.

Figure R3. Projected density of states (PDOS) of (a) Pt₁, (b) Pt₁/S-Graphene, (c) Pt₃₈ and (d) Pt₃₈/S-Graphene.

(3) Still about charge transfer, there is a similar recent work [Germano et al., ACS Appl. Mater. Interface 2019, 11, 5661] that maybe could be cited.

Response: We have cited the suggested work in the revised manuscript.

(4) To finish, the small translucent balls in Figure 6 are very hard to see, I recommend changing their color. The "translucent cyan belt" is not cyan in the figure. And there is a typo ("ZEP") of zero-point energy (ZPE) in Methods section.

Response: Thanks for the reviewer's valuable comment here. The translucent balls in Figure 6 has been modified to make them easier to distinguish. The "translucent cyan belt" has been corrected to "translucent pink belt". The typo in the Methods section has also been corrected.

Reviewer #2 (Remarks to the Author):

In the field of heterogeneous catalysis, metal-support interaction plays a crucial role for controlling the activity, selectivity, and even stability. This manuscript describes an interesting metal-support interaction phenomenon between Pt and sulfur-doped carbon supports. The authors found that the electrons transfer from platinum single atoms to sulfur-doped carbons, but the carbon supports conversely donate electrons to Pt when the size is expanded to nanocluster. Both experimental and DFT simulation evidences support such metal size-dependent reversing phenomenon of electron transfer. Further, they found that the electron-enriched Pt nanoclusters were more active than electron-deficient Pt single atoms for catalyzing hydrogen evolution reaction. Such principle certainly could be applied to other metals and other catalysis reactions, for example, the heterogeneous hydrogenation/dehydrogenation catalysis processes. Overall, I enjoy reading these new findings about the metal size-dependent electronic metal-support interaction, and I would like to recommend the acceptance of this work for publication in Nature Communications, if the following technical issues can be addressed well.

Response: We appreciate the reviewer for the positive comments.

1. Although the authors have already tried to explain why the Pt nanoclusters have much better HER activity than Pt single atom, I would suggest that more detailed discussion should be supplied for this issue. As many studies have shown that single platinum atom catalysts with high oxidation state can catalyze HER effectively, the low activity of PtSA/S-C may can not be only associated to its electron deficiency.

Response: Thanks for the reviewer's valuable comment here. We agree that in addition to the influence of electronic structure, the chemical environment can also affect the catalytic activity of Pt, especially in single atom catalysts, similar to molecular catalysts. The superior HER activity of single platinum atom catalysts reported in literature could be owing to the positive effect of its coordination atoms, such as C, N, although these single platinum atom catalysts are electron-deficient. Recently, Chang

Hyuck Choi et al. also reported a S-doped carbon supported single platinum catalyst with poor alkaline HER activity (J. Am. Chem. Soc. 2018, 140, 16198-16205), which is consistent with our work, suggesting that the coordination environment does have significant effects on the catalytic activity of single atom catalysts.

The above discussion has been added in the revised manuscript (page 9).

2. DFT simulations showed that the enrichment of electrons on platinum atoms could reduce absorption energies of hydrogen atom and promote HER activity. The authors also claimed that the Pt nanoclusters supported on S-doped carbon is electron-enriched compared to the ones supported S-free carbon black BP2000. Therefore, more experiment results such as the activity comparison between S-doped carbon and BP2000 supported platinum catalysts are needed to support such viewpoint.

Response: Thanks for the reviewer's valuable comment here. The HER activity of S-doped carbon and BP2000 supported platinum catalysts has been compared and shown in Fig. R4. The S-doped carbon supported catalyst with higher electron density exhibits higher HER activity than BP2000 supported one. Such result is consistent with DFT analyses, that is, the enrichment of electrons on platinum atoms could reduce absorption energies of hydrogen atom and promote the HER activity at some hydrogen absorption sites.

The above data and related discussion have been added in the revised manuscript (page 9) and Supplementary information (Fig. S21).

Figure R4. HER activity of PtNC/BP2000-300 and PtNC/S-C-500 (without IR correct).

3. Considering that some chlorine residual was possibly left on the PtSA/S-C catalyst prepared at a relatively low temperature and negatively affect the HER activity, more characterizations are needed to confirm the complete leaving of chlorine from H₂PtCl₆, such as XPS and temperature-programmed reduction.

Response: Thanks for the reviewer's valuable comment here. As suggested by the reviewer, we additionally carried out the XPS and temperature-programmed reduction measurements to confirm the complete leaving of chlorine from the H₂PtCl₆ precursor. The XPS results of PtSA/S-C clearly indicate that there is no signal in typical regions of Cl 2p (Fig. R5). Temperature-programmed reduction results also show that the Pt species supported on S-C have been reduced before 200 °C (Fig. R6).

The above data and related discussion have been added in the revised manuscript (page 3) and Supplementary information (Fig. S4 and 5).

Figure R5. Cl 2p XPS of PtSA/S-C and PtNC/S-C.

Figure R6. Temperature-programmed reduction curve of H_2PtCl_6 on S-C.

4. In Figure 6c, not all platinum atoms containing more electrons show lower ΔG , the authors should make further explanation.

Response: The reviewer's valuable comments here inspired us to re-check the structure of Pt_{38}/S -Graphene; and we made some adjustments to obtain a more stable Pt_{38}/S -Graphene structure. The updated model is shown in Fig. R7, in which 12 hydrogen adsorption sites were tested (denoted as site1 to site12, respectively). The corresponding δG of these adsorption sites are marked in Fig. R7 as well. δG is defined as $\delta G = \Delta G(\text{Pt}_{38}/\text{S}\text{-Graphene}) - \Delta G(\text{freestanding Pt}_{38})$. All the involved ΔG are negative values, so $\delta G > 0$ means that the HER activity of the corresponding

adsorption site on Pt₃₈/S-Graphene is better than that of free-standing Pt₃₈ and the negative value comes to the contrary conclusion. There are five adsorption sites have negative δG values, while the δG value in the other seven adsorption sites is greater than or equal to zero.

We try to make further explanation about the δG by using COHP (J. Phys. Chem. 1993, 97, 8617-8624; J. Phys. Chem. A 2011, 115, 5461-5466; J. Comput. Chem. 2016, 37, 1030-1035). The negative and positive values of COHP represent the bonding interaction and anti-bond interaction between atoms, respectively. The COHP analysis results for two adsorption sites with negative δG (site2 and site6) and two adsorption sites with positive δG (site5 and site8) are shown in Fig. R8. The bonding interactions in the Pt-H bonds are mainly distributed in the range of -4 to -8 eV. At site2 and site6, the peak intensity of bonding interaction increases, meaning that the bonding interaction in the Pt-H bond is enhanced, resulting in an increase adsorption strength of hydrogen at these sites. However, at the site5 and site8 sites, the peak intensity of bonding interaction decreases, which means that the bonding interaction in the Pt-H bond is weakened, resulting in weakening of the adsorption strength of hydrogen at these adsorption sites and leading to a positive δG value.

Combing control experiments and DFT results on Pt(111) slab, we have concluded that the enrichment of electron on platinum can improve its HER activity at some hydrogen absorption sites. However, it should be noticed that the electron density is not the only influence factor for HER. The chemical environment and geometrical effect can also affect catalytic activity. More importantly, the adsorption site and adsorption mode would inevitably influence catalytic activity, especially for particle catalysts, where the edge and the defect would exhibit unanticipated activity. Therefore, it is reasonable that a fraction of adsorption sites do not show lower ΔG .

The above data and related discussion have been added in the revised manuscript (page 11) and Supplementary information (Fig. S26).

Figure R7. Difference bader charge analysis and difference ΔG energy analysis of Pt₃₈/S-Graphene. The value of Δe in Pt atom can be read from the color bar. 12 hydrogen absorption sites are denoted as site1 to site12, respectively. The small white balls with white or blue color number mark the position of absorption site with the corresponding δG .

Figure R8. Projected crystal orbital Hamilton populations (pCOHP) averaged over all Pt-H bonds: (a) site2; (b) site5; (c) site6; (d) site8. pCOHP of spin-up and spin-dn are almost the same. The red arrow marks the change in peak intensity of Pt₃₈/S-Graphene relative to freestanding Pt₃₈.

5. The charge transfer resistance (R_{ct}) of Pt/S-C and commercial Pt/C should be

further measured and compared to analyze the electron-transfer dynamics involved in HER.

Response: Thanks for the reviewer's valuable comment here. The Nyquist plots of Pt/S-C and commercial Pt/C were obtained at a potential of -0.24 V (versus Ag/AgCl) over a frequency range of 100 mHz ~ 100kHz (Fig. R9). The charge transfer resistance of PtNC/S-C is much smaller than the PtSA/S-C and comparable to Pt/C, demonstrating the fast electron-transfer rate and rapid HER kinetics.

The above data and related discussion have been added in the revised manuscript (page 9) and Supplementary information (Fig. S20).

Figure R9. Nyquist plots of Pt/S-C and commercial Pt/C.

6. Some low-magnification HAADF-STEM images should be supplied in the SI part to show the homogeneity of the PtNC/S-C catalysts.

Response: Thanks for the reviewer's valuable comment here. Additional low-magnification HAADF-STEM images (Fig. R10) have been added in the revised Supplementary information to verify the homogeneous dispersion of Pt clusters over the S-C support.

The above data and related discussion have been added in the revised manuscript (page 4) and Supplementary information (Fig. S6).

Figure R10. Low-magnification HAADF-STEM images of PtNC/S-C.

Reviewer #3 (Remarks to the Author):

In the manuscript, the authors prepare the sulfur-doped carbon supported Pt single atoms (PtSA/S-C) and nanoclusters (PtNC/S-C) for the HER. PtNC/S-C exhibits much higher catalytic activity than Pt SA/S-C. The activity is also relatively higher than the previously reported catalysts. The enhanced catalytic activity of PtNC/S-C is explained well by the charge transfer from S-C support to PtNC. These results are very interesting; however, the reviewer cannot find the novelty in the MSI phenomenon. The phenomenon found in this study seems to be similar to the previously reported ones, although the authors persist that the new electronic SMSI phenomenon is confirmed by this work:

1. The authors persist the charge transfer varied with Pt particle size, but the proposal is based on the results of two samples of PtSA/S-C and PtNC/C. The size dependent charge transfer has been reported in carbon supported Pd catalysts, which demonstrates the positively charged interfacial Pd layer and the negatively charged second layer. (Nat. Commun, 2017, 8, 340)

Response: We appreciate the reviewer for the valuable comments. We have carefully read the mentioned literature on the carbon supported Pd catalysts (Nat. Commun, 2017, 8, 340), in which the authors mainly studied the influence of carbon surface functionalization and graphitization on the charge distribution at the Pd-C interface as well as on the catalytic selectivity. Also, they studied the charge distribution of different Pd particles ranging from 38 atoms to 239 atoms by DFT calculations. Certainly, this work is very interesting, particularly on the charge re-distribution at the Pd-C interface. Although the mentioned work demonstrated the positively charged interfacial Pd layer and the negatively charged second layer, the overall charge density of Pd particles is positive, suggesting that the direction of charge transfer is independent of particle size, which can be estimated from Figure 7 in the main text. While in our work, we report the size-dependent charge transfer reversal in the Pt/S-C system, that is, the direction of charge transfer can be tuned by particle sizes ranged from single atom to nanoclusters.

Moreover, according to the reviewer's comments, we explored the charge transfer between S-C and Pt with Pt atom numbers ranging from 1 to 44 by DFT calculations. Bader charge analysis results of a series of S-Graphene supported Pt_n cluster were shown in Fig. R11. Clearly, the charge was transferred from the substrate to the clusters in all the Pt_n/S-Graphene (n>1). Especially in the Pt₃₈/S-Graphene structure, the Pt₃₈ cluster capture 0.66e from the substrate. Combined with the experimental results, we believe that the direction of charge transfer in larger clusters (n>44) would remain the same as in Pt₃₈/S-Graphene. We show for the first time that the direction of charge transfer across the metal-support interface is opposite between single atom and nanocluster, which is different from the mentioned work.

However, according to the reviewer's comments, we have deleted the "new electronic SMSI phenomenon" words from the revised manuscript to make our discussion more objective. We have also cited the mentioned work (Nat. Commun, 2017, 8, 340) in the revised manuscript.

The above theory results and related discussion have been added in the revised manuscript (page 7 and 8) and Supplementary information (Fig. S14).

Figure R11. Bader charge analysis results of a series of S-Graphene supported Pt_n cluster.

2. When the electronic state of Pt clusters is change by the MSI, the structure of Pt clusters (e.g. Pt-Pt bond length or angle, particle shape) should be changed. However, the Pt model structure seems to be not changed from the one without the interaction

(Fig. 4). Do the authors fix the PtNC structure in the calculation? If so, does such calculation give a reliable result? In addition, is there no possibility that the MSI affect the particle structure to enhance the HER activity by geometrical effect? The structural change by the MSI and its influence on the catalysis should be addressed, since they have been confirmed in the previous studies as the authors refer to them in the introduction section (line 32, p.1 – line 2, p.3).

Response: Many thanks for the reviewer's valuable comment here. We had fully relaxed Pt₃₈/S-Graphene without fixing the Pt₃₈ structure. There are 24 Pt-Pt bonds in the Pt₃₈ crystal faces, labeled as *bond_num1* to *bond_num24*, respectively. The bond length distribution of freestanding Pt₃₈ cluster and Pt₃₈/S-Graphene are showed in Fig. R12. All the *bond_num* are 2.67 Å in Pt₃₈, while the Pt-Pt *bond_num* length is between 2.63Å to 2.79Å in Pt₃₈/S-Graphene. Although there is a strong electronic interaction between Pt cluster and S-Graphene, the structure of Pt₃₈ cluster underwent only a slightly deformation and seems very closed to the free-standing Pt₃₈ cluster when it was located on the S-Graphene substrate.

We agree that both geometrical effect and charge transfer could affect the HER activity of Pt₃₈ cluster. Here, we further studied the possible influence of geometrical effect according to the reviewer's comment. The δG is divided into two parts: $\delta G = \delta G_{\text{geo}} + \delta G_{\text{charge}}$, where δG_{geo} and δG_{charge} represent the influence of geometrical effect and charge transfer on the activity of Pt₃₈/S-Graphene, respectively. δG_{geo} is defined as $\delta G_{\text{geo}} = \Delta G(\text{Pt}_{38}\text{-fix}) - \Delta G(\text{Pt}_{38})$, where Pt₃₈-fix is Pt₃₈ cluster by removing the substrate in Pt₃₈/S-Graphene and fixing all Pt atoms during structure relaxation. δG_{charge} is defined as $\delta G_{\text{charge}} = \Delta G(\text{Pt}_{38}/\text{S-Graphene}) - \Delta G(\text{Pt}_{38}\text{-fix})$. We tested 12 hydrogen adsorption sites (labeled as site1 to site12, respectively) and calculated the δG , δG_{geo} , and δG_{charge} (Fig R7 and Table R1). The δG_{geo} of site8 and site9 are as large as 0.19 and 0.12 eV, respectively, which means that the geometric effect will have also an influence on the HER activity of Pt₃₈/S-Graphene. However, it should be noted that the charge transfer would affect the electronic structure of the system (Angewandte Chemie International Edition, 2018, 57, 12216-12226), and then induce the structural

deformation. The structural deformation can in turn affect the charge transfer from the substrate to the Pt cluster.

The above theory results and related discussion have been added in the revised manuscript (page 11) and Supplementary Information (Tab. S4).

Figure R12. 24 Pt-Pt bond_num of freestanding Pt₃₈ cluster and Pt₃₈/S-Graphene.

Table R1. ΔG of Pt₃₈/S-Graphene, freestanding Pt₃₈ and Pt₃₈-fix as well as δG , δG_{geo} and δG_{charge} .

	$\Delta G/eV$	$\Delta G/eV$	$\Delta G/eV$	$\delta G/eV$	$\delta G_{geo}/eV$	$\delta G_{charge}/eV$
Absorption site	Pt ₃₈ /S-Graphene	Freestanding Pt ₃₈	Pt ₃₈ -fix	Geometrical effect +charge	Geometrical effect	Charge
Site1	-0.51	-0.57	-0.55	0.06	0.02	0.04
Site2	-0.57	-0.53	-0.56	-0.04	-0.03	-0.01
Site3	-0.44	-0.52	-0.49	0.08	0.03	0.05
Site4	-0.59	-0.57	-0.55	-0.02	0.02	-0.04
Site5	-0.07	-0.53	-0.45	0.46	0.08	0.38
Site6	-0.58	-0.53	-0.53	-0.05	0.00	-0.05
Site7	-0.27	-0.28	-0.32	0.01	-0.04	0.05
Site8	-0.20	-0.57	-0.39	0.37	0.19	0.18
Site9	-0.22	-0.35	-0.23	0.14	0.12	0.01
Site10	-0.37	-0.31	-0.33	-0.05	-0.02	-0.03
Site11	-0.28	-0.27	-0.22	0.00	0.05	-0.06
Site12	-0.33	-0.27	-0.29	-0.06	-0.02	-0.04

Other comments:

- About the PtS₂ reference sample: The formal charge of Pt in PtS₂ would be 4+; however, the XANES white line of PtS₂ is much smaller than PtO₂, and similar to Pt foil rather than PtO₂. Is the XAFS spectrum of PtS₂ reliable?

Response: Many thanks for the reviewer's valuable comment here. We re-checked the data quality of XAFS of the PtS₂ sample that was purchased from Sigma-Aldrich. The *k* space of *k*²-weighted Pt L₃-edge of PtS₂ shown in Fig. R14 shows the high data quality with only minor noise when *k* is larger than 11 Å⁻¹, indicating the reliability of the PtS₂ XAFS spectrum. We noted that such difference of white line intensity has also occurred in the reported work (The Journal of Physical Chemistry C 2012, 116, 25790-25796), although the author did not explain this issue there. We assume that such difference could be attributed to the stronger oxidability of oxygen than sulfur, which impairs the electron density of Pt 4d orbit and lowers the white line intensity of Pt.

- In order to guarantee the quality of the XAFS data, it is better to show the EXAFS oscillations before Fourier transformation in the supporting information.

Response: Many thanks for the reviewer's valuable comment here. The *k* space of *k*²-weighted Pt L₃-edge of PtSA/S-C, PtNC/S-C, PtO₂, PtS₂ and Pt foil in Fig. R13 has been added in the Supplementary Information (Fig. S9), supporting the high data quality of all samples.

Figure R13. k space of k^2 -weighted Pt L_3 -edge of PtSA/S-C, PtNC/S-C, PtO₂, PtS₂ and Pt foil.

- The S2p XPS of S-C support is presented in supplementary figure 3; however, the spectra of supported Pt samples are not presented. S2p XPS or S K edge XAFS analysis of supported Pt samples would be helpful to confirm the Pt-S formation.

Response: Many thanks for the reviewer's valuable comment here. Actually, we had tried to confirm the Pt-S bond formation by S2p XPS, S K-edge and S L-edge XAFS (Figs R14-R16). However, since sulfur is multivalent and there are many S sites over the S-C remaining uncoordinated to Pt according to the EDS mapping of PtNC/S-C, we can not obtain useful and exact information about Pt-S bond or charge transfer. Nevertheless, by comparing the R space of Pt XAFS between PtNC/S-C, PtS₂, and PtO₂ (Figure 3), we can strongly verify the formation of Pt-S bond in PtNC/S-C.

Figure R14. S 2p XPS of Pt/S-C, S-C and PtS₂.

Figure R15. S K-edge of Pt/S-C and S-C.

Figure R16. S L-edge of Pt/S-C, S-C and PtS₂.

- Fig. 2: How is the (200) determined? The lattice might be (111).

Response: Many thanks for the reviewer's valuable comment here. As shown in Fig. R17, the measured interplanar spacing of PtNC/S-C is 0.2314 nm and 0.2066 nm respectively, close to the standard lattice spacing of (111) and (200) of platinum. Moreover, the measured intersection angle of two lattice planes in fast-Fourier transform (FFT) pattern is 120.57° , which is also close to the theoretical intersection angle 125.2° of (111) and (200). The deviation of 5° could arise from the drift of sample under STEM mode. If the marked (200) is assigned to (111), then considering the intersection angle, the lattice width of 0.2314 nm can only be ascribed to (200) or (220), which would result in great deviation of lattice spacing between measured sample and theoretical values.

Figure R17. (a) High resolution HAADF-STEM image of PtNC/S-C. (b) Fast-Fourier transform (FFT) pattern of (a).

Reviewer #1 (Remarks to the Author):

I recommend the acceptance of this work for publication in Nature Communications as it is.

Response: We sincerely appreciate the reviewer for the comments.

Reviewer #2 (Remarks to the Author):

The authors have made great efforts to perform additional experimental and computational works to address my concerns. They made further discussion about the poor activity of PtSA/S-C. Also, they performed control experiments to support the DFT results that the electron enrichment on PtNC/S-C can improve the HER activity. Other technical issues have also been addressed well. I enjoy reading the interesting findings demonstrated in this work, particularly about the metal size-dependent charge transfer reversal in the S-C supported Pt catalysts. I therefore recommend the acceptance of this work now for publication in Nature Communications.

Response: We sincerely appreciate the reviewer for the comments.

Reviewer #3 (Remarks to the Author):

All issues have been solved. In my opinion, this manuscript can be accepted for publication.

Response: We sincerely appreciate the reviewer for the comments.